# DelAC: A Multi-agent Reinforcement Learning of Team-Symmetric Stochastic Games

## Abstract

In this paper we study team-symmetric games with two teams. Players within the same team have common payoff functions. We show that team-symmetric games always have a team-symmetric Nash equilibrium. We develop and solve a linear complementarity problem of team-symmetric Nash equilibria. We propose an actor-critic based multi-agent reinforcement learning algorithm for team-symmetric games. Through simulations, we show that this multi-agent reinforcement learning algorithm performs much better than many existing algorithms.

**Keywords**: symmetric games, team games, reinforcement learning, multi-agent, Nash equilibrium, actor-critic

## 1 Introduction

For a long time games have been a versatile tool to model various types of interactions among multiple entities including collaboration and competition. Multi-agent reinforcement learning (MARL) emerges as a powerful tool to train agents to learn their equilibrium strategies. There is a rich literature on MARL (Albrecht et al., 2024a). In an MARL model, interactions among agents are typically governed by a stochastic game. One of the most widely accepted solution concepts of games is the Nash equilibrium. Indeed, many early research results on MARL have focused on training agents to play according to Nash equilibria of the game as a goal (Shoham & Leyton-Brown, 2007) (Hu & Wellman, 2003a). However, recently (Daskalakis et al., 2006) and (Chen & Deng, 2006) showed that computing Nash equilibria is PPAD-complete. Thus, it is important to develop MARL methods for special games that demand less computational complexity, while important in practice.

In this paper we study the MARL algorithms of team symmetric games. Symmetric games and team games are special games that find many real-world applications, such as cooperative robots, autonomous vehicles, and etc. We refer the reader to (Djehiche et al., 2017) for applications in engineering, and (Sanchez & Doncel, 2024) for an application in epidemic models. Symmetric games refer to games in which players have indistinguishable identities (Nash, 1951)(Emmons et al., 2024). Team games usually refer to games in which players have common payoff functions (Emmons et al., 2024). Symmetric games have been widely applied to study problems in economics, biology, social sciences, and engineering. We refer the reader to (Lee, 2025) and (Hefti, 2017) for details. A closely related game is the anonymous games (Daskalakis & Papadimitriou, 2007) (Kalai, 2005). In this paper we study a symmetric game with two teams. Specifically, players in the same team have indistinguishable identities and have a common payoff function. This game can model cooperation within teams and competition between teams. We call this game a team-symmetric game. We show that team-symmetric games always have a team-symmetric Nash equilibrium. The linear complementarity problem for the team-symmetric Nash equilibria has much lower complexity. We develop a multi-agent reinforcement learning algorithm for a stochastic team-symmetric game. Our algorithm is based on actor-critic architecture and numerical solution of linear complementarity problem of the game. Through simulations we show that our algorithm performs much better than many existing MARL algorithms in the literature. We mention that theoretic results on symmetric games were explored in (Hefti, 2017) and (Lee, 2025). Hefti (2017) studied symmetric equilibria and asymmetric

equilibria of a symmetric games and pointed out a condition for asymmetric equilibria to exist.

The outline of this paper is as follows. In Section 2 we present team-symmetric games and show that such games always have symmetric Nash equilibria. In Section 3 we present a computation procedure for a symmetric Nash equilibrium of a team-symmetric game. In Section 4 we present DelAC algorithm. We present simulation results in Section 5 and conclusions of this paper in Section 6.

## 2 Team-Symmetric Games

We consider a stochastic normal-form game $\mathcal{G} = (\mathcal{N}, \mathcal{O}, \mathcal{A}, u, P, \gamma)$, where $\mathcal{N}$ is a set of players, $\mathcal{A}$ is a set of actions available to the players, $\mathcal{O}$ is a local observations, $u$ is payoffs, $P$ is a transition probability function, and $\gamma$ is a discount factor in $[0, 1)$. In this paper we assume that there are two teams. Team $i$ has $n_i$ players for $i = 1, 2$. We label the players such that $\mathcal{N}_1 = \{1, 2, \ldots, n_1\}$ and $\mathcal{N}_2 = \{n_1 + 1, n_1 + 2, \ldots, n_1 + n_2\}$. Thus, $\mathcal{N} = (\mathcal{N}_1, \mathcal{N}_2)$. We assume that players in the same team has the same action spaces. Let $\mathcal{A}_i$ be the action space of a player in team $i$. Thus, $\mathcal{A} = (\mathcal{A}_1)^{n_1} \times (\mathcal{A}_2)^{n_2}$ is the product of action spaces of players in the two teams. Without loss of generality, we assume that $\mathcal{A} = \{1, 2, \ldots, |\mathcal{A}|\}$. In this paper we use boldface letters to denote vectors and matrices. Let $\boldsymbol{u} = (u_1, u_2, \ldots, u_{n_1+n_2})$ be payoff functions, where $u_i : \mathcal{A} \to R$ is the payoff for each player $i$. Let $\Gamma(\boldsymbol{u})$ denote a normal-form game with payoff functions $\boldsymbol{u}$. We present a few definitions.

**Definition 1** *We say that $\mathcal{G}$ has common-payoff within teams if*

$$u_i(\boldsymbol{a}) = u_j(\boldsymbol{a}), \text{for all } i, j \in \mathcal{N}_1 \text{ and for all } i, j \in \mathcal{N}_2 \tag{1}$$

*and for all action profiles $\boldsymbol{a} \in \mathcal{A}$.*

**Definition 2** *A permutation within teams $\phi$ is a bijection from $\mathcal{N}_1 \cup \mathcal{N}_2$ to itself with additional properties that $\phi(i) \in \mathcal{N}_1$ for any $i \in \mathcal{N}_1$ and $\phi(j) \in \mathcal{N}_2$ for any $j \in \mathcal{N}_2$.*

**Definition 3** *Suppose that player $i$ takes action $a_i$. We call $\mathcal{G}$ a team-symmetric game if*

$$u_{\phi(i)}(a_1, a_2, \ldots, a_{n_1+n_2}) = u_i(a_{\phi(1)}, a_{\phi(2)}, \ldots, a_{\phi(n_1+n_2)}) \tag{2}$$

*for all $(a_1, a_2, \ldots, a_{n_1+n_2}) \in \mathcal{A}$, $1 \le i \le n_1 + n_2$.*

Note that (1) and (2) imply that

$$u_i(a_1, a_2, \ldots, a_{n_1+n_2}) = u_{\phi(i)}(a_{\phi(1)}, a_{\phi(2)}, \ldots, a_{\phi(n_1+n_2)}). \tag{3}$$

We represent mixed strategies by $|\mathcal{A}|$ dimensional real vectors. Specifically, let $\boldsymbol{v}_{ia}$ denote an $|\mathcal{A}|$ dimensional vector, whose entries are all zero, except that the $a$-th entry is one. $\boldsymbol{v}_{ia}$ denotes the pure strategy $a$ of player $i$. We use notation $\boldsymbol{s}_i$ to denote a mixed strategy of player $i$. The $a$-th entry of $\boldsymbol{s}_i$ is the probability that player $i$ uses action $a$. Mixed strategy $\boldsymbol{s}_i$ is a point in a simplex with vertices $\{\boldsymbol{v}_{ia}\}$. The expected payoff of player $i$ when strategy profile $\boldsymbol{s} = (\boldsymbol{s}_1, \boldsymbol{s}_2, \ldots, \boldsymbol{s}_{n_1+n_2})$ is used is

$$\sum_{\boldsymbol{s}} u_i(\boldsymbol{a}) \prod_{j=1}^{n_1+n_2} (\boldsymbol{s}_j)_{a_j}, \tag{4}$$

where $(\boldsymbol{s}_j)_{a_j}$ denotes the $(a_j)$-th entry of $\boldsymbol{s}_j$. We overload notations and use $u_i$ to also denote the expected payoff of player $i$. We adopt a standard notation that $\boldsymbol{s}_{-i} = (\boldsymbol{s}_1, \ldots, \boldsymbol{s}_{i-1}, \boldsymbol{s}_{i+1}, \ldots, \boldsymbol{s}_{n_1+n_2})$. Let $(\boldsymbol{s}_{-i}, \boldsymbol{s}')$ denote

$$(\boldsymbol{s}_1, \ldots, \boldsymbol{s}_{i-1}, \boldsymbol{s}', \boldsymbol{s}_{i+1}, \ldots, \boldsymbol{s}_{n_1+n_2}),$$

*i.e.* player $i$'s strategy in profile $\boldsymbol{s}$ is replaced by $\boldsymbol{s}'$ and other players' strategies are unchanged. Note that (3) can be extended to mixed strategies, *i.e.*

$$u_i(\boldsymbol{s}_1, \boldsymbol{s}_2, \ldots, \boldsymbol{s}_{n_1+n_2}) = u_{\phi(i)}(\boldsymbol{s}_{\phi(i)}, \boldsymbol{s}_{\phi(2)}, \ldots, \boldsymbol{s}_{\phi(n_1+n_2)}) \tag{5}$$

Also note that $\phi$ is a permutation within teams on players. It can induce a permutation on strategies. Specifically, define permutation $\rho$ such that

$$\rho(\boldsymbol{s}) = (\boldsymbol{s}_{\phi(i)}, \boldsymbol{s}_{\phi(2)}, \dots, \boldsymbol{s}_{\phi(n_1+n_2)}).$$

**Definition 4** *We call strategy profile $\boldsymbol{s}$ team symmetric or symmetric within teams if*

$$\rho(\boldsymbol{s}) = \boldsymbol{s} \tag{6}$$

*for all permutation $\rho$ within teams.*

One main result of this paper is the following proposition. Its proof is presented in Appendix.

**Proposition 5** *A team-symmetric game with common payoffs has a team-symmetric Nash equilibrium.*

## 3  Computation of team-symmetric Nash equilibria

In this section, we formulate a feasibility program to numerically compute a team symmetric Nash equilibrium of a game with two teams and common payoffs within teams. We consider strategy profile of the form

$$(\boldsymbol{s}, \boldsymbol{t}),$$

where $\boldsymbol{s}$ and $\boldsymbol{t}$ are strategy profiles used by members in team 1 and team 2, respectively. Since strategy profiles are symmetric within teams, we assume that

$$\boldsymbol{s} = (\boldsymbol{x}, \boldsymbol{x}, \dots, \boldsymbol{x}),$$
$$\boldsymbol{t} = (\boldsymbol{y}, \boldsymbol{y}, \dots, \boldsymbol{y}),$$

where $\boldsymbol{x}$ (resp. $\boldsymbol{y}$) is a mixed strategy adopted by players in team 1 (resp. team 2). Since we assume that game $\mathcal{G}$ is team symmetric and has common payoffs, we rewrite payoff functions so that they are functions of number of players who adopt specific actions. Let

$$\hat{u}_i(\boldsymbol{g}, \boldsymbol{h})$$

be the common payoff function of players in team $i$, where $i = 1, 2$. The $j$-th entry of vector $\boldsymbol{g}$ (resp. $\boldsymbol{h}$), $g_j$, denotes the number of players in team 1 (resp. team 2) who adopt action $j$, $1 \le j \le |\mathcal{A}|$. Additionally, we introduce notation

$$\hat{u}_{i,a}(\boldsymbol{g}, \boldsymbol{h}), \tag{7}$$

which is the payoff of team $i$ when one of the player in team $i$ adopts action $a$, and vectors $\boldsymbol{g}$ and $\boldsymbol{h}$ contain the numbers of players adopting actions in the two teams. In (7), vectors $\boldsymbol{g}$ and $\boldsymbol{h}$ do not include the player in team $i$ who takes action $a$. Assume that $i = 1$. Since players sample actions independently according to their mixed strategies, vector $\boldsymbol{g}$ (resp. $\boldsymbol{h}$) occur independently according to multinomial distributions with parameters $n_1 - 1$ and $\boldsymbol{x}$ (resp. $n_2$ and $\boldsymbol{y}$). The expected payoff of team 1 as a function of team strategies $\boldsymbol{x}$ and $\boldsymbol{y}$ is

$$\tilde{u}_{1,a}(\boldsymbol{x}, \boldsymbol{y}) = \sum_{\substack{\boldsymbol{g} \\ \sum_j g_j = n_1 - 1}} \sum_{\substack{\boldsymbol{h} \\ \sum_j h_j = n_2}} \left[ \frac{(n_1 - 1)!}{\prod_{j=1}^{n_1} g_j!} \prod_{j=1}^{|\mathcal{A}|} x_j^{g_j} \cdot \frac{n_2!}{\prod_{j=1}^{n_2} h_j!} \prod_{j=1}^{|\mathcal{A}|} y_j^{h_j} \cdot u_1(\boldsymbol{g} + \boldsymbol{e}_a, \boldsymbol{h}) \right], \tag{8}$$

where $\boldsymbol{e}_a$ is vector whose entries are all zero, except that the $a$-th entry is one. We mention that in a special case where

$$u_1(\boldsymbol{g}, \boldsymbol{h}) = \sum_{j=1}^{|\mathcal{A}|} c_{1,j} g_j + \sum_{k=1}^{|\mathcal{A}|} c_{2,k} h_k$$

for some constants $\{c_{1,j} : 1 \le j \le |\mathcal{A}|\}$ and $\{c_{2,k} : 1 \le k \le |\mathcal{A}|\}$, $\tilde{u}_{1,a}$ can be further simplified as

$$\tilde{u}_{1,a}(\boldsymbol{x}, \boldsymbol{y}) = c_{1,a} + \sum_{j=1}^{|\mathcal{A}|} (n_1 - 1) c_{1,j} x_j + \sum_{k=1}^{|\mathcal{A}|} n_2 c_{2,k} y_k.$$

The expected payoff of players in team 2 is similar.

It is well known that Nash equilibria can be numerically obtained by solving a linear complementarity problem (LCP) (Shoham & Leyton-Brown, 2009). This LCP can be viewed as a nonlinear program without objective functions, *i.e.* a feasibility program. The feasibility program for the team-symmetric games with common payoffs is significantly simpler than that of a typical general-sum game. We present the program for the sake of completeness.

$$\tilde{u}_{1,j}(\boldsymbol{x},\boldsymbol{y}) + r_{1,j} \leq U_{1,j}^*, \qquad 1 \leq j \leq |\mathcal{A}|, \tag{9}$$

$$\tilde{u}_{2,k}(\boldsymbol{x},\boldsymbol{y}) + r_{2,k} \leq U_{2,k}^*, \qquad 1 \leq k \leq |\mathcal{A}|,$$

$$\sum_{j=1}^{|\mathcal{A}|} x_j = 1, \quad \sum_{k=1}^{|\mathcal{A}|} y_k = 1,$$

$$x_j \geq 0, \quad r_{1,j} \geq 0, \quad x_j \cdot r_{1,j} = 0, \qquad 1 \leq j \leq |\mathcal{A}|,$$

$$y_k \geq 0, \quad r_{1,k} \geq 0, \quad y_k \cdot r_{2,k} = 0, \qquad 1 \leq k \leq |\mathcal{A}|.$$

## 4 Delegate Actor-Critic Networks

In this section we present our reinforcement learning algorithm to compute a symmetric Nash equilibrium of a team-symmetric stochastic game. Actor-critic algorithms form a paradigm for multiagent reinforcement learning of stochastic games. In this framework, an actor is a neural network that trains a parameterized policy based on policy gradient algorithms. In addition, a neural network called critic is introduced in this frame to estimate the state-value of the game. We further remark that actor-critic methods follow a centralized training and distributed execution (CTDE) paradigm.

We first describe a baseline algorithm. We then simplify the algorithm. The simplified version of the algorithm will be called delegate actor-critic (DelAC) algorithm. We allocate one actor and one critic to each player. Initially, all actors generate the same mixed strategy distributions, and all critic networks generate zero $Q$ values. At time $t$, the actor associated with player $i$ generates a mixed strategy for player $i$, where $1 \leq i \leq n_1 + n_2$. Player $i$ samples an action $a_i^t$ from this mixed strategy. An action profile $\boldsymbol{a}^t = (a_1^t, a_2^t, \ldots, a_{n_1+n_2}^t)$ is executed, and the environment generates observations $\boldsymbol{o}^{t+1}$ and rewards $\boldsymbol{u}^t$. The discounted reward of player $i$ from time $t$ onward is defined as

$$d_i^t = \sum_{\tau=t}^{\infty} \gamma^{\tau-t} u_i^\tau, \tag{10}$$

where $\gamma$ is a discount factor in interval $(0,1)$, and $u_i^t$ is the reward received by player $i$ at time $t$. We remark that the superscript of $\gamma^\tau$ in (10) denotes powers, and the superscripts in $d_i^t$ and $u_i^t$ indicate time. We also remark that we overload the symbol $u$ here. The action-value function $Q_i(s,a)$ for player $i$,

$$Q_i^\pi(s,a) = E\left[d_i^t | s^t = s, a_i^t = a\right], \tag{11}$$

is the expected discounted reward when selecting action $a$ in state $s$ and following policy $\pi$.

Critic networks are trained to generate action-value functions using $\{(\boldsymbol{o}^\tau, \boldsymbol{o}^{\tau+1}, \boldsymbol{a}^\tau, \boldsymbol{u}^\tau), 1 \leq \tau \leq t\}$. Overloading symbol $Q$, we denote the output of critic $i$ by $Q_i(\boldsymbol{o}^t, \boldsymbol{a}^t)$ in order to emphasize that it is trained with $\{(\boldsymbol{o}^\tau, \boldsymbol{a}^\tau), 1 \leq \tau \leq t\}$. We also emphasize that all critics are trained with the same data and in the same order. We construct a normal-form game $\Gamma(\boldsymbol{Q}(\boldsymbol{o}^t, \boldsymbol{a}^t))$, in which payoffs of player $i$ is $Q_i(\boldsymbol{o}^t, \boldsymbol{a}^t)$ for $1 \leq i \leq n_1 + n_2$. According to Proposition 6 to be presented below, game $\Gamma(\boldsymbol{Q}(\boldsymbol{o}^t, \boldsymbol{a}^t))$ is a team-symmetric game. We solve a Nash equilibrium of the game using (9). Denote the solution of (9) by $\{\tilde{\pi}_i^t, 1 \leq i \leq n_1 + n_2\}$. The actor network of player $i$ attempts to learn $\tilde{\pi}_i^t$ by minimizing loss function

$$\mathcal{L}_i(t) = \sum_{\tau=1}^{t} D_{\mathrm{KL}}\left(\tilde{\pi}_i^\tau \| \pi_i(\cdot \mid o_i^t)\right), \tag{12}$$

where $D_{\text{KL}}(\cdot)$ denotes the KL divergence.

The actor associated with player $i$ attempts to learn player $i$'s mixed strategy in a symmetric Nash equilibrium within teams. The $i$-th critic attempts to learn the action-value function $Q_i^\pi(s,a)$ of player $i$. The $i$-th critic learns $Q_i(s,a)$ by minimizing loss function

$$L(t) = \sum_{\tau=1}^{t} \sum_{i=1}^{n_1+n_2} \left( Q_i(\boldsymbol{o}^\tau, \boldsymbol{a}^\tau) - y_i^\tau \right)^2,$$

where $y_i^t$ is a sample value of $d_i^t$ in (10).

The proof of Proposition 6 is quite simple and is presented in the Appendix.

**Proposition 6**      *1. Actor networks produce symmetric mixed strategy distributions within teams, i.e. $\pi_1(\cdot|o_1) = \phi_i(\cdot|o_i)$ for $1 \le i \le n_1$ and $\pi_{n_1+1}(\cdot|o_{n_1+1}) = \phi_i(\cdot|o_i)$ for $n_1 + 1 \le i \le n_1 + n_2$.*

     *2. The normal-form game $\Gamma(\boldsymbol{Q}(\boldsymbol{o}^t, \boldsymbol{a}^t))$ produced by the critic is symmetric within teams, i.e. $Q_1(\boldsymbol{o}^t, \boldsymbol{a}^t) = Q_i(\boldsymbol{o}^t, \boldsymbol{a}^t)$ for $1 \le i \le n_1$ and $Q_{n_1+1}(\boldsymbol{o}^t, \boldsymbol{a}^t) = Q_i(\boldsymbol{o}^t, \boldsymbol{a}^t)$ for $n_1 + 1 \le i \le n_1 + n_2$.*

Due to Proposition 6, each team needs only one delegate actor and one delegate critic. In fact, the two delegate critic networks can be merged into one network. We present the DAC algorithm in Algorithm 1. In Algorithm 1 we choose actors 1 and $n_1 + 1$ as the delegates of the two teams.

## 5   NUMERICAL AND SIMULATION RESULTS

In this section we present simulation of Algorithm 1 on several team-symmetric games. We compare the performance of Algorithm 1 with those of several well known MARL algorithms in the literature. We present a brief review of these baseline algorithms in Section 5.1. The performance of Algorithm 1 is presented in Section 5.2.

### 5.1   REVIEW OF SOME MADRL ALGORITHMS

In this section we review a few well known MARL algorithms, with which the performance of Algorithm 1 will be compared. These algorithms can be broadly classified into two categories, value based and actor-critic based methods. All baseline methods considered are compatible with the centralized training with decentralized execution (CTDE) paradigm, unless otherwise noted.

Independent Q-learning (IQL) is a value based algorithm which treat agents as independent learners (Mnih et al., 2013). Each agent independently determines his/her action without coordinating with other agents. Nash Q-learning (NashQ) (Hu & Wellman, 2003b) is another value based method. Nash Q-learning approximates payoffs of a stochastic game by Q-values and computes a corresponding Nash equilibrium. Friend-or-Foe Q-learning (FFQ) classifies agents into a set of friends and a set of foes. FFQ applies a min-max operation to all foes and a max-sum operation to friends (Littman et al., 2001). QMIX (Rashid et al., 2018) is a value decomposition method developed for cooperative settings. It factorizes a centralized action-value function into individual per-agent Q-functions using a monotonic mixing network. NWQMIX (Wu, 2024) extends QMIX to environments where both cooperation and competition exist. For cooperative agents, original factorization proposed by QMIX is retained. For competing agents, new factorization is proposed.

For actor-critic based methods, we compare Algorithm 1 with Independent Advantage Actor-Critic (IA2C) (Lowe et al., 2017), Independent Proximal Policy Optimization (IPPO) (Schulman et al., 2017), (Papoudakis et al., 2020), Centralized Advantage Actor-Critic (CA2C) (Lyu et al., 2023), and Multi-Agent Proximal Policy Optimization (MAPPO) (Yu et al., 2022). IA2C allocates an actor and a critic to each agent. Actors and critics of all agents learn independently without centralized coordination. As IA2C,

---

**Algorithm 1** Delegate Actor-Critic Network

---

1: Initialize delegate actor networks which generate $\pi_1(\cdot|\boldsymbol{o}^t, \boldsymbol{u}^t)$ and $\pi_{n_1+1}(\cdot|\boldsymbol{o}^t, \boldsymbol{u}^t)$
2: Initialize centralized critic which generates $Q(\boldsymbol{o}^t, \boldsymbol{u}^t)$
3: **for** each episode = 1 to $M$ **do**
4:   Obtain initial observations $(o_1, \ldots, o_N)$
5:   **for** each time step $t = 1$ to $T$ **do**
6:     **for** each agent $i = 1$ to $n_1 + n_2$ **do**
7:       Sample action $a_i^t \sim \pi_1(\cdot|\boldsymbol{o}^t, \boldsymbol{u}^t)$ for $1 \le i \le n_1$
8:       Sample action $a_i^t \sim \pi_{n_1+1}(\cdot|\boldsymbol{o}^t, \boldsymbol{u}^t)$ for $n_1 + 1 \le i \le n_1 + n_2$
9:     **end for**
10:    Form joint action $\boldsymbol{a}^t = (a_1^t, \ldots, a_{n_1+n_2}^t)$
11:    Execute $\boldsymbol{a}^t$
12:    Receive payoffs $\boldsymbol{u}^t$ and observations $\boldsymbol{o}^{t+1}$ of the next time step
13:    Store $(\boldsymbol{o}^t, \boldsymbol{u}^t, \boldsymbol{a}^t, \boldsymbol{o}^{t+1})$ into a batch
14:   **end for**
15:   Convert each joint action profile $\boldsymbol{a}^t$ in a batch into team-wise action count vectors $(\boldsymbol{g}^t, \boldsymbol{h}^t)$ as defined in (7)
16:   Compute targets for each time step in a batch:

$$y_j^t = \sum_{k=t}^{T} \gamma^{k-t} u_j^k, \quad \text{for } j = 1, 2 \tag{13}$$

17:   Update critic by minimizing total loss over batch:

$$L(\theta) = \sum_t \sum_{j=1}^{2} \left( Q_j(\boldsymbol{o}^t, \boldsymbol{a}^t) - y_j^t \right)^2$$

18:   **for** each $t$ in batch **do**
19:     Construct game $\Gamma(\boldsymbol{Q}(\boldsymbol{o}^t, \boldsymbol{a}^t))$ from the critic
20:     Solve (9) for equilibrium $\tilde{\pi}^t$ from $\Gamma(\boldsymbol{Q}(\boldsymbol{o}^t, \boldsymbol{a}^t))$
21:     **for** each agent $i = 1$ to $n_1 + n_2$ **do**
22:       Accumulate KL divergence loss in (12)
23:     **end for**
24:   **end for**
25:   **for** each agent $i = 1$ to $n_1 + n_2$ **do**
26:     Update actor $\phi_i$ using gradient descent on $\mathcal{L}_i$
27:   **end for**
28: **end for**

---

IPPO allocates an actor and a critic to all agents, and these actors and critics perform proximal optimization independently. CA2C incorporates a centralized critic that has access to global state and joint action information during training. See Chapter 9.4.3 in (Albrecht et al., 2024b). Both CA2C and MAPPO follow the CTDE paradigm.

We remark that in the implementation of the MARL methods above, we adopt a parameter sharing architecture (Terry et al., 2020) and (Christianos et al., 2021). Agents within the same team share the same value network and policy network. Despite sharing parameters, agents retain independent observations and actions, allowing for decentralized execution during both training and evaluation.

## 5.2 Results

We simulate 30 randomly generated two-team symmetric games. In these games, each team has two players and each player has two actions. The payoffs are randomly selected integers in the range $[0, 10]$. We compare the average mean squared error (MSE) of DelAC and the MARL algorithms in Section 5.1. In each time step, we compute the average of the two team MSEs. A team MSE is defined as an MSE between agents' expected payoffs and the expected payoffs corresponding to a symmetric Nash equilibrium. Fig. 1 contains the average MSE over 30 random team-symmetric zero-sum games. Fig. 2 contains the average MSE over 30 random team-symmetric general-sum games. From these two figures, we see that DelAC has the least average MSE compared with other methods.

We have also simulated a benchmark game called the generalized matching penny (GMP) game. The payoff matrix of this game is shown in Table 1. In a GMP, there are two teams and each team has two agents. Each agent has two actions, *i.e.* action H and action T. From Table 1, a GMP is a zero-sum game. According to (Kalogiannis et al., 2023), when the game parameter $\omega$ satisfies $0 < \omega < 1$, a GMP game admits a unique Nash equilibrium, which is a mixed strategy. This property eliminates the existence of pure strategy equilibria, thereby establishing GMP as a particularly challenging environment for evaluating convergence behaviors in mixed-strategy settings. Fig. 3 shows DelAC has least MSE compared with those of other methods.

From Figs. 1, 2 and 3 we see that DelAC consistently outperforms other methods. In the training process, DelAC maintains a continuously decreasing MSE, while other methods either converge to non-optimal values or exhibit significant swings.

|  | $HH$ | $HT/TH$ | $TT$ |
|---|---|---|---|
| $HH$ | $1, -1$ | $\omega, -\omega$ | $-1, 1$ |
| $HT/TH$ | $-\omega, \omega$ | $0, 0$ | $-\omega, \omega$ |
| $TT$ | $-1, 1$ | $\omega, -\omega$ | $1, -1$ |

Table 1: Payoff matrix of a GMP game.

We study the average KL divergence between learned strategy profiles and Nash equilibria. These averages are taken with respect to 30 random games and four players in each game. The results are shown in Table 2. We see that DelAC has much smaller average KL divergences than other methods.

|  | General-sum | Zero-sum | GMP($\omega$=0.5) |
|---|---|---|---|
| IA2C | 2.276±4.900 | 1.584±4.396 | 1.303±2.197 |
| IPPO | 3.343±7.231 | 2.379±6.722 | 2.591±3.563 |
| CA2C | 0.858±2.718 | 0.363±1.634 | 1.128±2.907 |
| MAPPO | 0.744±2.431 | 0.301±1.575 | 1.005±2.789 |
| DelAC | **0.078±0.563** | **0.001±0.001** | **0.001±0.004** |

Table 2: Average KL divergence between learned policies and Nash equilibria.

## 6 Conclusions

In this paper we studied team-symmetric games with two teams. Players within the same team have common payoff functions. We showed that team-symmetric games always have

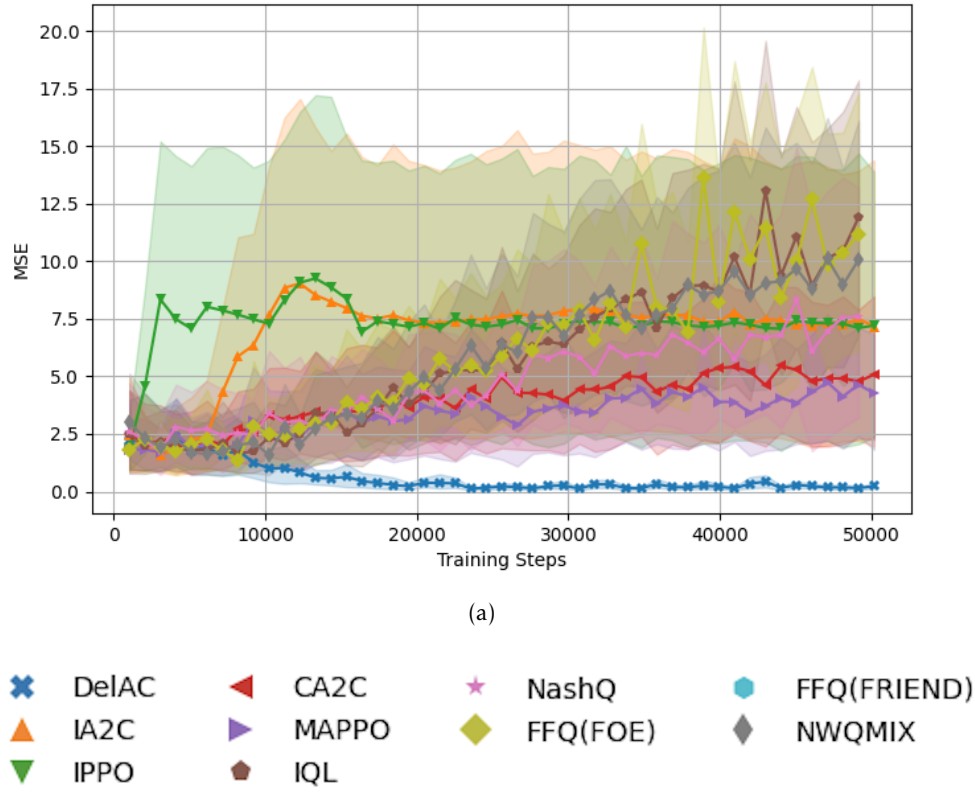

(a)

(b)

Figure 1: Panel (a) contains the average MSE of 30 random symmetric zero-sum games with two teams. Panel (b) contains curve legends of the curves in panel (a).

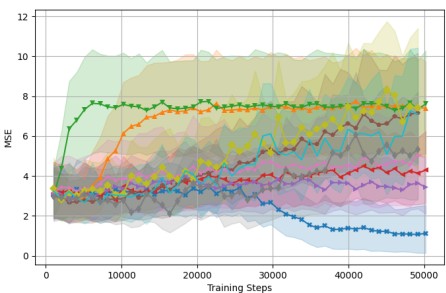

Figure 2: Average MSE of 30 random symmetric, general-sum games with two teams.

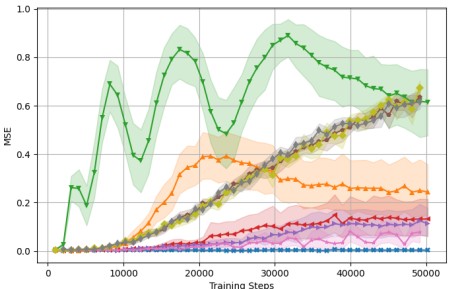

Figure 3: MSE of GMP game with $\omega$=0.5.

a team-symmetric Nash equilibrium. We developed and solve a linear complementarity problem of team-symmetric Nash equilibria. We proposed an actor-critic based multi-agent reinforcement learning algorithm for team-symmetric games. Through simulations, we showed that this multi-agent reinforcement learning algorithm performs much better than many existing algorithms.

## A  Appendix

In this appendix we present a proof of Proposition 5 and Proposition 6. The proof of Proposition 5 is quite similar to that of Theorem 2 in (Nash, 1951), except that the permutations involved in this paper are permutations within teams. The proof shall apply Brouwer's fixed point theorem in Chapter II of (Bredon, 2013). For completeness, we quote the theorem as follows.

**Theorem 7 (Brouwer's fixed point theorem)** *Every continuous function from a nonempty convex compact subset $K$ of a Euclidean space to $K$ itself has a fixed point.*

**Proof of Proposition 5.** Define function $f : R^{n_1+n_2} \to R^{n_1+n_2}$. Let

$$f(\boldsymbol{s}) = \hat{\boldsymbol{s}}, \tag{14}$$

where

$$\hat{\boldsymbol{s}}_i = \frac{\boldsymbol{s}_i + \sum_j \max\left(0, u_i(\boldsymbol{s}_{-i}, \boldsymbol{v}_{ij}) - u_i(\boldsymbol{s})\right)\boldsymbol{v}_{ij}}{1 + \sum_j \max\left(0, u_i(\boldsymbol{s}_{-i}, \boldsymbol{v}_{ij}) - u_i(\boldsymbol{s})\right)}. \tag{15}$$

It has been shown in the proof of Theorem 1 in (Nash, 1951) that $f$ has a fixed point and this fixed point is a Nash equilibrium point. We now present and prove the following two claims.

1. The set of team symmetric strategy profiles is a compact convex subset of the simplex with vertices

$$\left\{(\boldsymbol{s}_1,\ldots,\boldsymbol{s}_{n_1+n_2} : \sum_{j=1}^{|\mathcal{A}|}(\boldsymbol{s}_i)_j = 1, 1 \le i \le n_1 + n_2\right\}.$$

2. Function $f$ maps a team symmetric strategy profile into a team symmetric strategy profile.

It follows from the two claims and Brouwer's fixed point theorem that $f$ in (15) has a fixed point which is symmetric within teams.

It remains to prove the two claims. For claim 1, suppose that $\boldsymbol{s}$ and $\boldsymbol{t}$ are two team symmetric strategy profiles. Let $w$ be a real number in $[0,1]$. Then, for permutation $\phi$ within teams and its induced permutation $\rho$,

$$\rho(\boldsymbol{s}) = \boldsymbol{s}$$
$$\rho(\boldsymbol{t}) = \boldsymbol{t},$$

and

$$\rho(w\boldsymbol{s} + (1-w)\boldsymbol{t})$$
$$= \rho\left(w\boldsymbol{s}_1 + (1-w)\boldsymbol{t}_1,\ldots,w\boldsymbol{s}_{n_1+n_2} + (1-w)\boldsymbol{t}_{n_1+n_2}\right)$$
$$= \left(w\boldsymbol{s}_{\phi(1)} + (1-w)\boldsymbol{t}_{\phi(1)},\ldots,w\boldsymbol{s}_{\phi(n_1+n_2)} + (1-w)\boldsymbol{t}_{\phi(n_1+n_2)}\right)$$
$$= w\rho(\boldsymbol{s}) + (1-w)\rho(\boldsymbol{t})$$
$$= w\boldsymbol{s} + (1-w)\boldsymbol{t}. \tag{16}$$

Thus, the set of team symmetric strategy profiles form a convex set. Let $\mathcal{S}$ be the set of team symmetric strategy profiles, *i.e.* $\mathcal{S}$ contains all $\boldsymbol{s}$ such that $\rho(\boldsymbol{s}) = \boldsymbol{s}$ for any permutation $\rho$

within teams. Endow $\mathcal{S}$ with a metric

$$d(\boldsymbol{s}, \boldsymbol{t}) = \sum_{i=1}^{n_1+n_2} \mathrm{TV}(\boldsymbol{s}_i, \boldsymbol{t}_i),$$

where TV is the total variation distance between two probability mass functions. Let $\mathcal{S}^c$ be its complement. Suppose that $\boldsymbol{s} \in \mathcal{S}^c$. There must be at least two players, say $i$ and $j$, such that $\boldsymbol{s}_i \neq \boldsymbol{s}_j$. Define

$$\mathbf{N}(\boldsymbol{s}) = \Big\{ \boldsymbol{t} : d(\boldsymbol{s}, \boldsymbol{t}) < \mathrm{TV}(\boldsymbol{s}_i, \boldsymbol{s}_j) \Big\}.$$

It is clear that $\rho(\boldsymbol{t}) \neq \boldsymbol{t}$ for any $\boldsymbol{t} \in \mathbf{N}(\boldsymbol{s})$. Thus, $\mathbf{N}(\boldsymbol{s}) \subset \mathcal{S}^c$. It follows that $\boldsymbol{s}$ is an interior point, and thus, $\mathcal{S}^c$ is open.

Now we prove claim 2. Suppose that $\boldsymbol{s}$ is a team symmetric strategy profile, *i.e.*

$$\rho(\boldsymbol{s}) = \boldsymbol{s}. \tag{17}$$

Let $\phi$ be a permutation within teams and let $\rho$ be its induced permutation on strategy profiles. Suppose that $\phi(i) = k$. Then, the $i$-th entry of $f(\boldsymbol{s})$ is given in (15). Since the $k$-th entry of $\rho(\boldsymbol{s})$ is $\hat{\boldsymbol{s}}_i$, it is clear that the $k$-th entry of $f(\rho(\boldsymbol{s}))$ is

$$(f(\rho(\boldsymbol{s})))_k = \hat{\boldsymbol{s}}_i.$$

Thus,

$$f(\rho(\boldsymbol{s})) = \rho(f(\boldsymbol{s})). \tag{18}$$

We have

$$f(\rho(\boldsymbol{s})) = f(\boldsymbol{s}) = \hat{\boldsymbol{s}} = \rho(\hat{\boldsymbol{s}}), \tag{19}$$

where the first equality in (19) is due to (17), the second equality is due to (14) and the third equality is due to (18). Thus, $\hat{\boldsymbol{s}}$ is team symmetric.

Next, we prove Proposition 6. The proof is quite straightforward, and is based on induction.

**Proof of Proposition 6.** Initially at $t = 1$ all actors generate the same mixed strategy distributions, and all critic networks generate zero $Q$ values. Thus, the two statements of the proposition hold. Assume that the two statements hold for some $t$. Since the back-propagation training of all critic networks in the same team uses the same data and in the same order, it follows that all critic networks in the same team are identical at time $t + 1$. Thus, statement two of the proposition is true at time $t + 1$. Since the normal-form game at time $t$ has symmetric payoffs within teams, it follows that the game possess a symmetric Nash equilibrium. Since the actor networks are symmetric within teams and are trained by a symmetric strategy profile, it follows that at time $t + 1$ the actors remain symmetric within teams. Thus, statement one holds at time $t + 1$. We thus, complete the induction procedure.

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
