# OpenReview forum: "DelAC: A Multi-agent Reinforcement Learning of Team-Symmetric Stochastic Games"
_ICLR.cc/2026/Conference — Submitted to ICLR 2026_

### Official Review · Reviewer_9Bva · 2025-10-23

**Soundness:** 2
**Presentation:** 2
**Contribution:** 1
**Rating:** 2
**Confidence:** 3

**Summary:**

This paper studies a team-symmetric two-team game in which members within each team share the same payoff. The authors prove the existence of a Nash equilibrium for this setting and propose an actor–critic multi-agent reinforcement learning algorithm to approximate it. They then evaluate the method on 30 randomly generated games. However, the paper has substantial limitations in three areas: the discussion of related work, the methodological/theoretical contribution, and the experimental validation. Overall, I do not find the contribution sufficient for ICLR.

**Strengths:**

1. The paper writing is easy to follow.

2. The proposed method is simple and easy to reproduce.

**Weaknesses:**

1. Literature review:  The related-work section is incomplete. Stochastic games have a long history, and several influential MARL algorithms for stochastic/Markov games are not discussed. Please broaden the survey and position the contribution relative to key prior art (see, e.g., Yang et al., “Game-theoretic Multi-Agent Reinforcement Learning,” arXiv:2011.00583, 2025).

2. Theoretical properties: The proposed method is simple, but its theoretical guarantees are unclear. Given the paper’s emphasis on equilibrium, can you establish convergence (or convergence under assumptions) to a Nash equilibrium, or at least provide conditions under which the method approximates an equilibrium? Negative results or counterexamples would also be informative.

3. Experimental evaluation: The evaluation on 30 randomly generated two-team symmetric games is not fully convincing. Please consider more established benchmarks and stronger baselines, along with ablations and robustness analyses, to demonstrate generality and practical relevance.

[1] Yang Y., Ma C., Ding Z., et al. Game-theoretic Multi-Agent Reinforcement Learning. arXiv:2011.00583, 2025.

**Questions:**

Please see the weaknesses.

---

### Official Review · Reviewer_n96g · 2025-10-27

**Soundness:** 3
**Presentation:** 1
**Contribution:** 1
**Rating:** 2
**Confidence:** 3

**Summary:**

This paper investigates the setting of team-symmetric games, where two opposing teams are composed of identical, interchangeable agents with the same payoffs. First, they prove that such games always admit a team-symmetric equilibrium, that is a strategy profile where each agent within the same team plays the same (possibly mixed) strategy. Then they propose a practical multi-agent reinforcement learning algorithm, based on actor-critic, to solve these problems.

**Strengths:**

The investigated setting is interesting: game theory is one of the basis for multi-agent reinforcement learning, and the foundation for many practical advancements in real-life problems, and thus both theoretical and practical advancements are very much welcomed.

**Weaknesses:**

The proposed MARL algorithm does not sound novel: to me it looks only like an actor-critic structure using parameter sharing amongst players in the same team. The only difference between existing actor-critic algorithms and the proposed DelAC is that in the latter we are not following the policy gradients to optimize the (shared) policy, but rather we are solving exactly a game with action-values as the payoffs, and then minimizing a KL-divergence term between our current solution and the optimum we found. This approach leads to scalability issues: with more agents, actions and observations involved in the original game, solving precisely the game induced by the $Q$-function will become quickly impractical.

Another point is the clarity of the paper: notation is very sloppy and difficult to follow precisely, making it quite difficult to get a rigorous understanding of the theoretical definitions and contributions of the paper. Please see the Questions below for some more detailed comments on this.

**Questions:**

- Why do we have $u_{\phi(i)}$ on the l.h.s. of Equation (2) and not $u_i$? Indeed, combining Equations (1) and (2) we obtain Equation (3), as you point out.

- If we defined $\mathcal{A}=(\mathcal{A})^{n_1}\times (\mathcal{A})^{n_2}$ as the set of joint actions/pure strategies for all players, then why we represent a mixed strategy for player $i$ as an $|\mathcal{A}|$-dimensional vector? This would be an exponentially big vector, with one entry for each possible joint pure strategy. Your definition of $\mathbf{v}_{ia}$ looks wrong: how can it represent the pure strategy $a$ for player $i$? To me it looks like representing the joint pure strategy $a$ for the whole player set. The correct definition would be using $|\mathcal{A}_i$-dimensional vector; with this all of the following definitions would work fine...

- In Equation (4), the summation should be over $\mathbf{a}\in\mathcal{A}$ and not over $\mathcal{S}$ no?
- What are $U_{1,j}^\*$ and $U_{2,k}^\*$ in Equation (9)?

- In Equation (11), you are mentioning states $s$, but you never introduced those. Indeed, agent $i$ is only receiving its local observation $o_i^t$ from the environment in the context you presented. Also, you never introduced or even mentioned the concept of policy $\pi$ before this point.

- Your proposed MARL method does not seem to be using any policy gradient at all: you learn an action-value function $\mathcal{Q}_i(\mathbf{o}^{\tau},\mathbf{a}^{\tau})$, and you use it to build a normal-form game whose solution is the optimal decentralized policy for every agent, and finally you learn an approximation for this by minimizing a KL-divergence. Where are the policy gradients used here? This has the form of an actor-critic algorithm, but it does not seem like it is learned in the same way... Moreover, I do not see why it is called *delegate* actor-critic: it just look like a normal actor-critic structure (other than for the point discussed above) using parameter sharing amongst homogeneous cooperating agents, a very popular technique in cooperative MARL.

- In Proposition 6, why is $\pi_1(\cdot|o_1)=\phi_i(\cdot|o_i)$? $\phi$ was the permutation over the agents no? Why it is now the (shared) policy?

- In Algorithm 1, why do we now have both the policies $\pi_{\{1,n_1+1\}}(\cdot|\mathbf{o}^t,\mathbf{u}^t)$ and the critic $Q(\mathbf{o}^t,\mathbf{u}^t)$ conditioning on the payoff values $\mathbf{u)}^t$ in lines 1 and 2? Moreover, why do we need to convert the joint action $\mathbf{a}^t$ into the two count vectors $(\mathbf{g}^t,\mathbf{h}^t)$ when these are never used afterwards?

- The experimental problems used for evaluation are quite small: only 2 agents per team with two actions each. This is not a sufficiently broad set of experiments to validate your proposed method. The results are a bit expected: you are solving a game perfectly and then do KL-divergence minimization, which is a much stronger learning signal to follow than learning through policy gradients like IA2C/IPPO and CA2C/MAPPO: this is much akin to a supervised regression problem, which in general is significantly simpler than a RL problem with all of its involved moving components.

- Perhaps I do not see it, but the cyan curve of FFQ(FRIEND) is missing from Figure 1?

---

### Official Review · Reviewer_M389 · 2025-11-01

**Soundness:** 2
**Presentation:** 1
**Contribution:** 2
**Rating:** 2
**Confidence:** 3

**Summary:**

The paper studies two-team, team-symmetric stochastic games with common payoffs. It claims that such games admit a team-symmetric Nash equilibrium and that it can be solved with a linear complementarity problem. It proposes Delegate Actor-Critic (DelAC), which forms a per-timestep normal-form game from a centralized critic, trains delegate actors toward the computed equilibrium via KL loss, and reports lower MSE and KL than standard MARL baselines on random symmetric games and a generalized matching-pennies benchmark.

**Strengths:**

The problem to exploit within-team symmetry to improve learning stability and convergence is of practical importance to the game theory community.

**Weaknesses:**

There are numerous issues with this paper that lead me to believe that it isn't ready for publication at ICLR:
- A main result of the paper is not actually novel. Proposition 5 states: "A team-symmetric game with common payoffs has a team-symmetric Nash equilibrium." but this is a direct extension of well established game-theory results [1,2,3]. The appendix explicitly states that the proof is quite similar to [1].
- No training details, hyperparameters, baseline details, or code are provided. The results of this paper are not reproducible. Details on how training was conducted and how baselines were setup need to be added for the paper to be ready for acceptance.
- Overall writing quality, clarity, and formatting is below conference standards.
- MSE to Nash equilibrium payoffs is a poor performance metric. A better fit is a regret-based metric like NashConv or exploitability since these directly quantify distance from equilibrium. MSE between learned payoffs and “equilibrium payoffs” doesn’t measure incentives to deviate, and can judge non-equilibrium strategies as “close to NE” simply because their payoffs happen to match.

[1] Nash, John. “Non-Cooperative Games.” Annals of Mathematics, vol. 54, no. 2, 1951, pp. 286–295.

[2] Osborne, Martin J., and Ariel Rubinstein. A Course in Game Theory. MIT Press, 1994. pp. 35-36

[3] Plan, Asaf. “Symmetry in n-player games.” Journal of Economic Theory, vol. 207, 2023, Article 105549.

**Questions:**

In the experiment result figures, what are "training steps"? If these are iterations of each algorithm, the x-axes are not comparable between algorithms. If these are environment steps, it should be labeled as such.

---

### Official Review · Reviewer_GjWG · 2025-11-03

**Soundness:** 3
**Presentation:** 2
**Contribution:** 2
**Rating:** 2
**Confidence:** 3

**Summary:**

The paper studies a subclass of stochastic games in which agents form two teams with identical payoffs within each team but potentially opposing objectives across teams. The authors first formalize team-symmetric games, proving the existence of a team-symmetric Nash equilibrium under this structure using an adaptation of Brouwer’s fixed-point theorem.

**Strengths:**

1. The paper focuses on an interesting niche of multi-agent RL (team-symmetric games) and novelly integrates a classical game-theoretic solution (Nash equilibrium via an LCP) into a modern actor-critic algorithm.

2. The theoretical contributions, which prove the existence of a team-symmetric Nash equilibrium and provide an efficient computation method, appear sound and form a solid foundation for the proposed DelAC algorithm.

**Weaknesses:**

1. Marginal Novelty: The work is confined to a very narrow and idealized setting (two identical teams), which restricts its applicability. The core concepts, like policy sharing, are not new, making the novelty incremental.

2. Weak Experimental Evaluation: The empirical results are limited to extremely simple "toy" domains. The algorithm's superior performance is not surprising given it's designed for these exact scenarios, and its scalability to more complex environments is unproven.

3. Lack of Generalizability: The approach relies heavily on strict symmetry assumptions that are rare in the real world. The paper fails to discuss the algorithm's convergence properties or the computational overhead of its equilibrium-solving step.

4. Insufficient Empirical Analysis: The experimental section lacks depth, providing no ablation studies or detailed analysis to explain why the algorithm works well. It focuses only on final performance metrics without offering deeper insights into the learning dynamics.

**Questions:**

Please see pros and cons

---

### Meta-Review · Area_Chair_xuR2 · 2026-01-06

**Summary:**

The paper proposed an actor-critic MARL algorithm for two-team symmetric games. The reviewers expressed concerns around the rigor and clarity of the paper, the veracity of the claimed contributions (existence of symmetric NE in symmetric 2-team games is known), the novelty of the proposed MARL algorithm, and missing related work.

**Reviewer Concerns:**

The reviewers expressed concerns around the rigor and clarity of the paper, the veracity of the claimed contributions (existence of symmetric NE in symmetric 2-team games is known), the novelty of the proposed MARL algorithm, and missing related work.

**Reviewer Scores:**

No rebuttal was submitted so I do not expect the reviewers to have changed their scores.

---

### Decision · Program_Chairs · 2026-01-26

Reject